# Spironolactone for adult female acne (SAFA): protocol for a double-blind, placebo-controlled, phase III randomised study of spironolactone as systemic therapy for acne in adult women

Susanne Renz ![ORCID],[1] Fay Chinnery,[1] Beth Stuart,[1,2] Laura Day,[1] Ingrid Muller ![ORCID],[2] Irene Soulsby,[3] Jacqui Nuttall,[1] Karen Thomas,[4] Kim Suzanne Thomas ![ORCID],[5] Tracey Sach,[6] Louise Stanton,[1] Matthew J Ridd ![ORCID],[7] Nick Francis ![ORCID],[2] Paul Little ![ORCID],[2] Zina Eminton,[1] Gareth Griffiths,[1] Alison M Layton,[8] Miriam Santer ![ORCID] [2]

For numbered affiliations see end of article.

**Correspondence to**
Dr Miriam Santer;
m.santer@soton.ac.uk

## ABSTRACT

**Introduction** Acne is one of the most common inflammatory skin diseases worldwide and can have significant psychosocial impact and cause permanent scarring. Spironolactone, a potassium-sparing diuretic, has antiandrogenic properties, potentially reducing sebum production and hyperkeratinisation in acne-prone follicles. Dermatologists have prescribed spironolactone for acne in women for over 30 years, but robust clinical study data are lacking. This study seeks to evaluate whether spironolactone is clinically effective and cost-effective in treating acne in women.

**Methods and analysis** Women (≥18 years) with persistent facial acne requiring systemic therapy are randomised to receive one tablet per day of 50 mg spironolactone or a matched placebo until week 6, increasing to up to two tablets per day (total of 100 mg spironolactone or matched placebo) until week 24, along with usual topical therapy if desired. Study treatment stops at week 24; participants are informed of their treatment allocation and enter an unblinded observational follow-up period for up to 6 months (up to week 52 after baseline). Primary outcome is the Acne-specific Quality of Life (Acne-QoL) symptom subscale score at week 12. Secondary outcomes include Acne-QoL total and subscales; participant acne self-assessment recorded on a 6-point Likert scale at 6, 12, 24 weeks and up to 52 weeks; Investigator's Global Assessment at weeks 6 and 12; cost and cost effectiveness are assessed over 24 weeks. Aiming to detect a group difference of 2 points on the Acne-QoL symptom subscale (SD 5.8, effect size 0.35), allowing for 20% loss to follow-up, gives a sample size of 398 participants.

**Ethics and dissemination** This protocol was approved by Wales Research Ethics Committee (18/WA/0420). Follow-up to be completed in early 2022. Findings will be disseminated to participants, peer-reviewed journals, networks and patient groups, on social media, on the study website and the Southampton Clinical Trials Unit website to maximise impact.

**Trial registration number** ISRCTN12892056;Pre-results.

### Strengths and limitations of this study

► Pragmatic design to inform real-world decision making for women with acne includes a primary outcome that is a participant-reported outcome measure, broad eligibility and recruitment strategies via primary care, secondary care, community and social media advertising.

► Randomisation to either spironolactone or matched placebo, with participants in both groups using topical treatments as usual (creams, gels and lotions), if desired, in order to reflect the place of oral treatments in the acne care pathway.

► Adaptions during the COVID-19 pandemic included inevitable limitations, including remote follow-up visits (via phone or video call), limiting collection of secondary outcomes such as investigator-assessed acne severity.

## BACKGROUND

Acne vulgaris (from hereon referred to as acne) is the eighth most common disease worldwide[1] and typically starts in adolescence with 15%–20% of people affected showing moderate or severe acne, often persisting into adulthood.[2] Facial scarring occurs in approximately 20% of people, but the main impact is social, with levels of psychological disability equivalent to those seen in conditions such as asthma and diabetes.[3 4] Incidence of acne in adult women is considerable and growing.[5–9]

First-line treatment for acne is topical treatments either alone or combination preparations, containing retinoids, benzoyl peroxide and/or antibiotics.[3] However, non-adherence to topical treatments is common, possibly because of the need to be used consistently for up to 8 weeks, and adverse reactions, such as stinging or redness, are common.[10] People therefore commonly seek second-line therapies, such as oral antibiotics, ethinylestradiol/cyproterone acetate (co-cyprindiol) or combined oral contraceptives.[3] In the UK, oral isotretinoin can be used under the supervision of a dermatologist for indications including severe, cystic, nodular or recalcitrant acne. Oral isotretinoin is highly effective, but is not suitable for all patients and is teratogenic,[2] therefore needing careful pregnancy prevention management.

A third of people who consult with acne are prescribed long courses of oral antibiotics.[3 11] However, acne is a disease of sebogenesis, and antibiotics have no effect on sebum production.[12 13] Furthermore, rising rates of antibiotic resistance mean non-antibiotic alternatives are needed.[14]

Spironolactone, a potassium-sparing diuretic, is widely used in the UK for indications including hypertension[15] and has been used off-licence for women with acne for ≥30 years due to its antiandrogenic properties. US guidelines suggest a role in the management of female acne.[1] Spironolactone is not used to treat acne in men because of its feminising side effects.[16]

There is limited evidence for the effectiveness of spironolactone for treating acne, and the need for evidence from randomised controlled trials in this patient group is acknowledged.[16]

When considering spironolactone as a potential alternative systemic therapy for acne, one study reported treatment success in women who had previously failed isotretinoin,[17] and a second study comparing the frequency with which participants switched drug treatment within 1 year of initiation demonstrated no significant difference between those taking spironolactone and those taking oral antibiotics for their acne,[18] implying they are equally tolerated by users. A further database study has shown that spironolactone may have superior drug usage survival compared with oral antibiotics for women with acne, giving a suggestion of greater perceived effectiveness and tolerability.[19]

A James Lind Alliance Priority Setting Partnership, funded by the National Institute for Health Research (NIHR), identified the need to establish the best way to manage acne in women who may or may not have underlying hormonal abnormalities.[20] This informed an NIHR commissioned call (16/13 persistent acne in adult women) for proposals to answer the research question 'What is the effectiveness of spironolactone in the treatment of moderate-severe persistent acne in adult women?'

This study aimed to answer whether spironolactone, in addition to standard topical therapy is able to improve acne-related quality of life in adult women with moderate–severe persistent facial acne compared to placebo plus standard topical therapy.

## METHODS
### Study design and setting

The study is a phase III, multicentre, double-blind, randomised superiority study, to investigate clinical and cost-effectiveness of spironolactone in the treatment of moderate or severe persistent facial acne in adult women compared with placebo, in addition to standard topical treatment. The design is pragmatic in order to have strong external validity and to inform real-world decision making for women with acne and their health professionals. Pragmatic design includes broad eligibility and recruitment strategies, a primary outcome that is relevant to participants, low intensity follow-up and an intention-to-treat (ITT) analysis. 'Moderate to severe acne', in the context of this study, is defined as acne that warrants treatment with oral antibiotics, as judged by the potential participant and study clinician.

Baseline and follow-up appointments are carried out by UK hospital dermatology centres in order to facilitate blood tests at baseline and clinical assessments. Participants continue on the allocated treatment (spironolactone or placebo) in combination with their usual topical treatment (if desired) for a total duration of 24 weeks with assessments at weeks 6 and 12. Primary outcome is assessed at week 12 with the patient-reported Acne-specific Quality of Life (Acne-QoL).[21 22] From week 12, participants in both groups may receive 'usual care' from their usual clinical team, including oral treatments (oral antibiotics and hormonal treatments), and from week 24, participants may receive isotretinoin, if the participant and study clinician feel the need for rescue treatment. Trial participants receive shopping vouchers at baseline, 6 and 12 weeks (total £40). At week 24, participants stop taking their study treatment, are informed of their treatment group allocation and enter an unblinded observational follow-up period for up to 6 months (up to week 52 after baseline). During this observational final follow-up period, participants can ask their general practitioner (GP) to be prescribed spironolactone for their acne if they wish or pursue other acne treatments as part of usual care. Figure 1 illustrates the patient pathway through the study with table 1 summarising the study procedures.

Adaptations during the COVID-19 pandemic included the option to hold follow-up visits at weeks 6 and 12 remotely (phone/video calls) and to post out the study tablets and questionnaires to participants. Participants were also given standardised instructions on how to photograph their face to submit as part of remote follow-up assessments. Baseline visits continued to be face-to-face due to the requirement of a urine pregnancy test and blood test to assess kidney function and serum potassium levels.

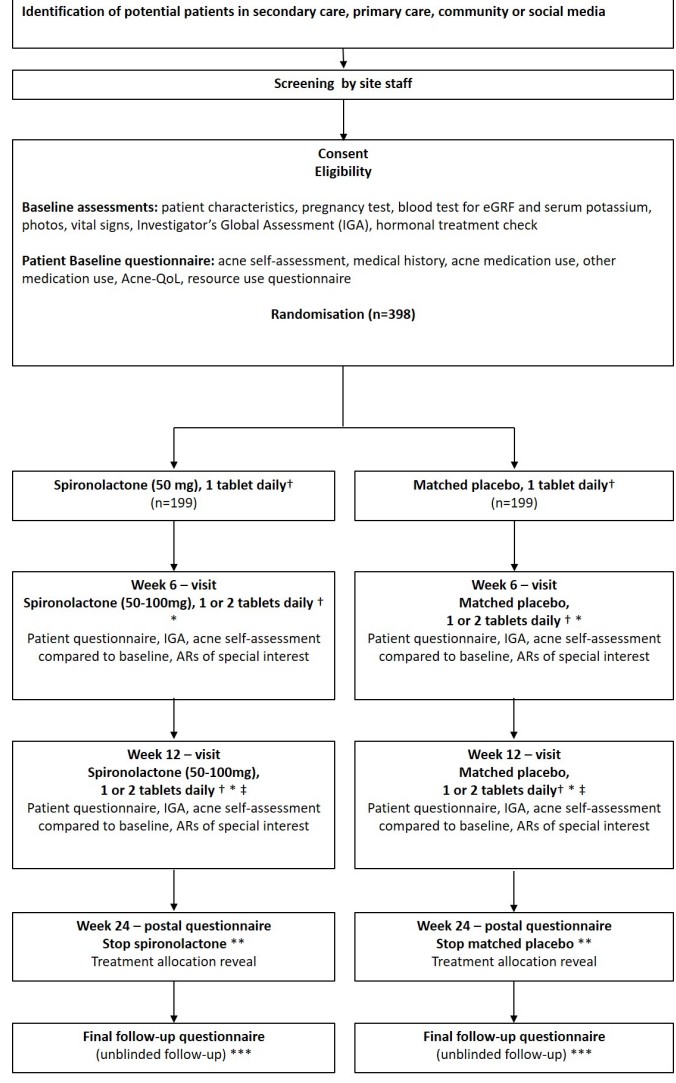

† Allow use of topical therapy (creams/lotions/gels)
*Escalate dose if study tablet is tolerated, otherwise maintain on 1 tablet
‡ Add antibiotic taken by mouth/change topical therapy if response to study tablet is inadequate
**Participants in either arm may seek to use spironolactone after this time.
***The follow-up questionnaire will be sent out 6 months or sooner after the week 24 timepoint.

**Figure 1** Study schem. AR, adverse reactions,

## Eligibility
### Inclusion criteria
Participants should fulfil all the following criteria:
► Women aged ≥18 years.
► Facial acne vulgaris, symptoms present for at least 6 months.
► Acne of sufficient severity to warrant treatment with oral antibiotics, as judged by the study clinician. Patients with an Investigator's Global Assessment (IGA) ≥2 are eligible to participate in the study.
► Women of childbearing potential at risk of pregnancy must be willing to use their usual hormonal or barrier method of contraception for the first 6 months of the study (while taking the study investigational medicinal product) and for at least 4 weeks afterwards.
► Willing to be randomised to either study group.
► Willing and able to give informed consent.

► Sufficient English to carry out primary outcome Acne-QoL.

### Exclusion criteria
Individuals meeting any of the following criteria will be excluded:
► Acne grades 0–1 using IGA (ie, clear or almost clear).
► Has ever taken spironolactone.
► Oral antibiotic treatment (lasting longer than a week) for acne within the past month.
► Oral isotretinoin treatment within the past 6 months.
► Started, stopped or changed long-term (lasting more than 2 weeks) hormonal contraception, ethinylestradiol/cyproterone acetate (co-cyprindiol) or other hormonal treatment within the past 3 months.
► Planning to start, stop or change long-term (lasting more than 2 weeks) hormonal contraception, ethinylestradiol/cyproterone acetate (co-cyprindiol) or other hormonal treatment within the next 3 months.
► Pregnant/breastfeeding.
► Intending to become pregnant in the next 6 months.
► Contraindicated to spironolactone:
  Currently taking potassium-sparing diuretic, ACE inhibitor, angiotensin II receptor blocker or digoxin.
  Hereditary problems of galactose intolerance, lactase deficiency or glucose–galactose malabsorption (as the spironolactone and placebo tablets contain lactose).
  Androgen-secreting adrenal or ovarian tumour.
  Cushing's syndrome.
  Congenital adrenal hyperplasia.
  Estimated glomerular filtration rate below 60 mL/min/1.73 m$^2$.
  Serum potassium level above the upper limit of reference range for the laboratory processing the sample.

### Intervention and control
Study participants receive one tablet per day (50 mg spironolactone or matched placebo) for the first 6 weeks of the study. At or any time after the week 6 visit, the dose is escalated to two tablets per day (total 100 mg spironolactone or matched placebo) by the study clinician, providing the participant is tolerating any side effects (see box 1). Participants are instructed to take their total dose once per day in the morning to avoid diuresis later in the day/evening. All known adverse effects of spironolactone are detailed in the patient information sheet, and trial participants have the opportunity to discuss the trial with the site team and discuss any questions before consenting to enter the trial.

Participants may use their usual topical treatments throughout the study, but adherence to topicals is not actively promoted as this may mask differences between the randomised groups. Participants are discouraged from changing their topical treatments between baseline and week 12.

**Table 1** Schedule of procedures

| Observation/procedure | Person undertaking the specified event | Visit | | | | |
| --- | --- | --- | --- | --- | --- | --- |
| | | Screening/ baseline | Week 6 | Week 12 | End of treatment/ week 24 contact | End of study/final follow-up questionnaire (52 weeks or sooner)* |
| **Enrolment** | | | | | | |
| Informed consent | Nurse/other cliniciant† | X | | | | |
| Eligibility evaluation | Other clinician | X | | | | |
| Participant characteristics | Nurse/other clinician | X | | | | |
| Blood pressure | Nurse/other clinician | X | | | | |
| Blood tests (serum potassium, eGFR) | Nurse/other clinician | X | | | | |
| Pregnancy test | Nurse/other clinician | X | | | | |
| **Randomisation** | Nurse/other clinician | X | | | | |
| **Assessments** | | | | | | |
| IGA | Nurse/other clinician | X | X | X | | |
| Medical history | Participant | X | | | | |
| Self-assessment—Participant's Global Assessment (adapted IGA) | Participant | X | X | X | X | X |
| Acne medication use | Nurse/other clinician/ participant | X | X | X | X | X |
| Other medication use | Nurse/other clinician/ participant | X | | | | |
| Acne-QoL | Participant | X | X | X | X | X |
| EQ-5D-5L | Participant | X | X | X | X | X |
| Resource use questionnaire | Participant | X | X | X | X | X |
| Self-assessment—comparison with baseline photo—6-point Likert Scale | Participant | | X | X | X | X |

Continued

**Table 1** Continued

| Observation/procedure | Person undertaking the specified event | Visit | | | | |
|---|---|---|---|---|---|---|
| | | Screening/baseline | Week 6 | Week 12 | End of treatment/week 24 contact | End of study/final follow-up questionnaire (52 weeks or sooner)* |
| Collection of ARs of special interest / Headache / Dizziness/vertigo/light headedness / Tingling / Indigestion/heartburn/dyspepsia / Diarrhoea / Polyuria (passing much more urine than normal) / Nausea/feeling sick / Vomiting/being sick / Tenderness of the breasts / Breast enlargement / Irregular menstrual periods / Abdominal pain / Weight gain / Reduced libido (reduced interest in sex) / Fatigue/tiredness / Drowsiness/sleepiness | Participant/other clinician | | X | X | X | |
| Serious adverse events | Other clinician (PI or delegate) | | X | X | X | |
| Assessment of treatment response to determine dose adjustment‡ | Participant/other clinician | | X | X | | |
| Satisfaction with study treatment | Participant | | | | X | |
| **Other activities** | | | | | | |
| Discuss use of contraception | Nurse/other clinician | X | X | X | | |
| Photographs of face taken | Nurse/other clinician | X | | | | |
| Photographs given to participant | Nurse/other clinician | X | | X (if a set was stored at site) | | |
| Letter to participant's GP (patient participation) | Nurse/other clinician | X | | | | |
| Check participant is not using oral acne treatment | Nurse/other clinician/participant | | X | X | | |
| Return excess IMP to clinic | Participant | | X | X | X (return via post) | |
| Spironolactone/placebo pill count | Nurse/other clinician/participant | | X | X | X | |
| Letter to participant's GP (if dose is changed) | Nurse/other clinician | | X | X | | |
| Reminder to participant to report any subsequent adverse event(s) that might reasonably be related to participation in this study (up to 52 weeks) | Nurse/other clinician | | | X | | |

Continued

## Table 1 Continued

| Observation/procedure | Person undertaking the specified event | Visit | | | | | |
|---|---|---|---|---|---|---|---|
| | | Screening/ baseline | Week 6 | Week 12 | End of treatment/ week 24 contact | End of study/final follow-up questionnaire (52 weeks or sooner)* |
| Ask participants if they would like to receive a summary of the study results, when available | Nurse/other clinician | | | X | | |
| Letter to participant (unblinding) | | | | | X (after 24 weeks) | |
| Letter to participant's GP (unblinding) | | | | | X (after 24 weeks) | |

*The follow-up questionnaire will be sent out 6 months or sooner after the 24-week time point.
†Dermatologist or clinical research fellow, in line with local procedures with demonstrable and appropriate level of training. Specific duties delegated by the PI and listed on the delegation log.
‡Dose escalated to two tablets per day if participant is tolerating side effects.
Acne-QoL, Acne-specific Quality of Life; AR, adverse reactions; eGFR, estimated glomerular filtration rate; GP, general practitioner; IGA, Investigator's Global Assessment; IMP, investigational medicinal product; PI, Principal Investigator.

---

### Box 1 Rationale for the dosing regimen

We conducted a survey of health professionals to inform the spironolactone dose regimen (unpublished). Responses were received from 41 dermatology consultants, 10 dermatology nurses and 3 dermatology Specialist Registrars .

Of these 54 dermatology health professionals, 22 prescribed spironolactone (9 rarely, 10 sometimes and 3 often). Most of those who responded stated that they would start at 50 mg and increase up to 100–150 mg, depending on response. Several noted that this would depend on the patient's weight, with the starting dose lowered to 25 mg if needed and allowing the dosage to increase up to 200 mg. There was no consistency on the time frame for these increases with 4, 6, 12 and 6 months all being mentioned as review points.

A previous Health Technology Assessment study examining common treatments in the management of acne suggested that assessing efficacy at 6 weeks was ideal[25]—this informed the timing of follow-up assessments and dose escalation. US guidelines note that studies have been carried out using spironolactone doses ranging from 50 mg/day to 200 mg/day.[1] No specific dose is recommended, but it is noted that side effects are dose-related.[1]

A recent hybrid systematic review of RCTs and case series identified some very low-quality evidence which showed that a daily dose of 200 mg was statistically significantly more effective than placebo versus inflamed lesions, but it also confirmed that this dose is associated with a significantly greater risk of adverse side effects than lower doses.[17]

Hence, there would appear to be no merit in using these higher doses for managing acne. Data from the multiple case series suggested that any future RCT examining lower doses is likely to generate results that confirm the effectiveness and better safety profile of doses of ≤100 mg/day, which informed the dosage regimen.

For most licensed indications for spironolactone, the British National Formulary states a starting dose of 100 mg, titrated as required. Therefore, a starting dose of 50 mg in the Spironolactone for Adult Female Acne study seems conservative.

### Intervention adherence

The hospital study team assesses participant adherence to treatment at each study visit using pill counts. In cases where only remote visits are possible, the participant informs the study team of the number of tablets remaining. However, no additional support or activity is undertaken to encourage daily pill taking, in keeping with the study's pragmatic design.

### Randomisation and assignment to intervention group

Following consent, participants are randomised 1:1 by clinical staff at the hospital site using an independent web-based system (TENALEA) using varying blocks of size 2 and 4, stratified by centre and by baseline severity (IGA <3 vs 3 or more) online supplemental file 1. The allocation sequence was generated by a statistician using computer-generated random numbers. Participants, study staff and investigators are blind to the treatment allocation until the end of the treatment period at week 24. Treatment allocation is revealed prior to week 24 only if there is a clinical need to do so.

## Recruitment

Potential participants, identified in primary and secondary care, direct advertising in areas close to recruiting hospitals and social media advertising, are directed to the study website (https://www.southampton.ac.uk/safa) or to contact the local study team directly.

In primary care, recruitment is supported by general practices acting as participant identification centres local to the recruiting centres identifying potential participants either opportunistically or via database search based on an acne diagnosis and relevant prescription within the past 6 months and mail-out of invitation pack. In secondary care, potential participants are identified opportunistically in outpatient clinics and through screening new referral letters.

We use targeted social media advertising to promote the study, build study awareness and interest.

Participants are free to withdraw consent from the study at any time without providing a reason. They may withdraw from study treatment but remain in follow-up; withdraw from study and follow-up, but give permission for their data to be used in analyses; or completely withdraw from the study and not permit their data to be used.

## Primary and secondary outcome measures

Clinically, the effectiveness of acne treatments is usually judged at 8–12 weeks, so the primary outcome in the study is assessed at week 12 with the Acne-QoL symptom subscale score.[21 22] The Acne-QoL was developed and validated for use in a clinical study to assess the impact of therapy on quality of life among people with facial acne, and the primary outcome at week 12 is the symptom subscale score of the Acne-QoL, because the Acne-QoL was intended to be presented as four separate subscale scores. It was not designed or validated to have a total score; however, it has published a minimum clinically important difference (MCID) of 2 points for the subscales and range 0–30 for symptom subscale score.[21–23] Other participant-reported outcome measures in acne do not have a published MCID available and have not been found to have advantages in terms of acceptability and validity.[24]

Secondary outcomes include Acne-QoL at week 24, participant self-assessed overall improvement recorded on a 6-point Likert Scale with photographs taken at baseline,[25] IGA,[5] Participant's Global Assessment, use of acne medication and participant satisfaction with study treatment. Trial participants are asked at each time point which topical or oral treatments were used in the period since the previous time point (if any) and have the option to add more information of the treatments for their acne in a free text box of the participant questionnaires. The IGA is a 5-point scale ranging from clear to severe (0, 'clear'; 1, 'almost clear'; 2, 'mild'; 3, 'moderate'; 4, 'severe').[6 26] The IGA[5 25] is used to grade the participant's acne as lesion counts are time-consuming, with wide interassessor variation and give little additional information to global assessments. All outcome measures are shown in table 2.

**Table 2** Schedule of observations

| Outcome measure | 6 weeks | 12 weeks (primary endpoint) | 24 weeks (end of treatment) | Unblinded follow-up (52 weeks or sooner)‡‡ |
|---|---|---|---|---|
| **Primary outcome measure** | | | | |
| Acne-QoL symptom subscale score | | X | | |
| **Secondary outcome measures** | | | | |
| Acne-QoL symptom subscale score | X | X | X | X |
| Acne-QoL other subscales | X | X | X | X |
| Acne-QoL total score* | X | X | X | X |
| Participant self-assessed overall improvement† | X | X | X | X |
| IGA‡ | X | X | | |
| Participant's Global Assessment§ | X | X | X | X |
| Participant satisfaction with study treatment¶ | | | X | |
| Health-related quality of life using EQ-5D-5L** | X | X | X | X |
| Costs incurred | X | X | X | X |
| Cost-effectiveness†† | | | X | |

*Self-perception, role—emotional and role—social.
†Recorded on a 6-point Likert Scale with photographs taken at the baseline visit to aid recall.[37]
‡5-point scale ranging from clear to severe (0, clear; 1, almost clear; 2, mild; 3, moderate; 4, severe).[25]
§Same scale as the IGA but written in plain English for participants' use.
¶Asked prior to revealing treatment allocation after 24 weeks.
**The EQ-5D-5 L assesses five dimensions: mobility, self-care, usual activities, pain/discomfort and anxiety/depression.
††Using EQ-5D-5L and data on health resource use during the study.
‡‡The follow-up questionnaire will be sent out 6 months or sooner after the 24-week time point.
IGA, Investigator's Global Assessment.

The safety profile of spironolactone is well established.[15 16] Consequently, we collect information about adverse reactions of special interest, both to inform the dose review decision from week 6 onwards as well as to learn more about incidence of side effects in this population. We also collect and report all serious adverse events (SAEs).

## Pregnancy

Spironolactone is considered contraindicated in pregnancy or a category C drug (ie, potential benefits may warrant use in pregnant women despite potential risks).[1 15] The main concern is around possible feminisation of the male fetus in the third trimester of pregnancy.[1] Women of childbearing potential at risk of pregnancy will be asked to use their usual hormonal or barrier method of contraception during the first 24 weeks of the study and for at least 4 weeks (approximately one menstrual cycle) afterwards. A pregnancy test will be conducted for all participants at their baseline visit and documented in their medical notes. At weeks 6 and 12, the study nurse/doctor will ask participants to confirm that they are still using contraception and have not changed their contraceptive method. Participants who become pregnant will be asked to inform their local site study team as soon as possible and will not be able to continue in the study.

## Sample size

Based on comparison of the Acne-QoL symptom score between groups at week 12, power 90%, alpha 0.05 and seeking a difference between groups of 2 points on the symptom subscale (SD 5.8, effect size 0.35), 346 participants are needed. Allowing for 20% loss to follow-up gives a total 434 participants (217 per group). Following discussions with oversight committees postfunding award, the sample size was recalculated. Allowing for a correlation with baseline of 0.293 and a deflation factor of $1-\rho2$[27] gives a total sample size required of 398 participants. A difference of 2 points on the symptom subscale and an SD of 5.8 (equivalent to an effect size 0.35) is in line with that reported in studies in a similar patient group and with the MCID reported for Acne-QoL.[21 22]

## Data collection methods

Participant data are entered into study electronic case report forms (eCRFs) via a remote data collection tool (Medidata Rave) by trained hospital research personnel with specific roles on the study and are regularly checked for missing or anomalous values by the Southampton Clinical Trials Unit (CTU) study staff.

## Data management

Participant data are pseudonymised by assigning each participant a participant identifier code, which is used to identify the participant during the study and for any participant-specific communication between Southampton CTU and recruiting sites. The site retains a participant identification code list which is only available to site staff and stored in a secure location at site.

## Patient and public involvement (PPI)

This study addresses a priority area identified as important to patients and health professionals by the James Lind Alliance Priority Setting Partnership for Acne.[7] We gained feedback from a virtual acne-specific patient panel, convened through 'People in Research' (https://www.peopleinresearch.org). A patient survey was carried out with the support of the UK Dermatology Clinical Trials Network in order to inform the study design. Findings suggested that participants would find it difficult to abstain from using topical treatments for more than 12 weeks and that asking participants to take a placebo for 1 year would also be a barrier to recruitment.

Two public contributors with experience of acne attend all Trial Management Group (TMG) meetings to ensure that decisions around the study design are informed by their perspective; study procedures are feasible for participants, and study materials are readable and include all the relevant information that participants would want. Public contributors influenced the trial design and delivery, for instance, by advocating use of social media advertising to improve recruitment, arguing against repeated measures in this patient group and that an upper age limit of 50 years was arbitrary and could appear discriminatory.

## Statistical methods

The study will be reported in accordance with Consolidated Standards of Reporting Trials (CONSORT) guidelines. A detailed statistical analysis plan (SAP) will be written and reviewed prior to the study database being locked.

The modified ITT population consists of all participants who have consented and have been randomised to a treatment arm and have complete data for the outcome being analysed. Analyses will be performed according to the modified ITT principle using a linear regression model. All analyses will be carried out in the modified ITT population, with the level of missing data reported, unless otherwise stated. The frequency and pattern of missing data will be examined and a multiple imputation model will be used as a sensitivity analysis if appropriate.

For the primary analyses, descriptive statistics will be obtained for the randomised groups to characterise recruited participants and assess baseline comparability. For the primary outcome, a linear regression model will be used to analyse Acne-QoL symptom subscale at week 12, adjusting for baseline variables (including baseline Acne-QoL symptom subscale score, use of topical treatments, use of hormonal contraception/ethinylestradiol/cyproterone acetate (co-cyprindiol) and randomisation stratification variables (centre, baseline severity (IGA <3 vs 3 or more)). A full list of covariates and model specification will be set out in the SAP. A 95% CI for the least squares mean difference between arms in Acne-QoL symptom subscale at week 12 will be calculated.

The same analysis methods will be used to summarise Acne-QoL symptom subscale at other time points (weeks 6, 24 and up to week 52 after baseline) and for the other

Acne-QoL subscales (self-perception, role—emotional and role—social) and total score. IGA and participants' comparison with baseline photo at weeks 6, 12 and 24 will be dichotomised as success or failure as recommended by the US Food and Drug Administration (with success for IGA and Participant's Global Assessment defined as clear or almost clear (grade 0 or 1) and at least a two-grade improvement from baseline; this represents a clinically meaningful outcome). The dichotomised outcomes will be summarised by frequencies and percentages and compared by group using logistic regression, adjusting for baseline assessment, use of hormonal contraception/ethinylestradiol/cyproterone acetate (co-cyprindiol), use of topical treatment and randomisation stratification factors.

Adverse reactions of special interest and SAEs will be summarised by group with frequencies and percentages and compared with Pearson's χ² tests. Logistic regression modelling will also be used to adjust for any important differences in topical treatment use by group. Subgroup analyses will investigate how the treatment effect differs by whether participants have symptoms consistent with polycystic ovary syndrome as recorded at the baseline visit. It is acknowledged the study is not powered for this subgroup analysis. The same analysis methods will be applied to the outcomes collected at up to 52 weeks; however, the interpretation of these results will be assessed with caution as participants will potentially have been off treatment for up to 6 months or have started a different acne treatment. All analyses will be carried out using Stata and/or SAS.

There are no planned interim analyses or subgroup analyses.

## Health economics analysis

If the intervention is found effective, a within-study economic evaluation will be undertaken to assess value for money of spironolactone plus usual care versus placebo plus usual care. The main perspective of the analysis will be that of the NHS over the 24-week treatment period, although a secondary analysis will assess the importance of a broader perspective by incorporating out-of-pocket costs related to acne and any productivity/employment impacts for people with persistent acne.[28]

Costs, including intervention and wider NHS resource use, are being recorded in the study eCRF for the former, while wider NHS resource use is captured in participant questionnaires at baseline and weeks 6, 12 and 24.

Costs will be valued using published unit costs for a common recent price year to estimate mean cost per participant in each arm.[29–31]

Review of the reliability, validity and responsiveness of three generic preference-based measures (EQ-5D, Short-Form Six Dimensions (SF-6D) and Health Utilities Index (HUI)) in skin conditions only found evidence to support the use of the EQ-5D in skin diseases with no studies looking at measurement properties for the SF-6D or HUI in skin disease.[32] Problems on the EQ-5D domains were found to be substantially higher in the acne sample receiving specialist care than in an age-truncated population sample (aged 20–39 years) particularly on the pain/discomfort (42.1% in the acne sample vs 17.7% in an age-truncated population sample) and anxiety/depression domains (52.8% vs 12.5%, respectively).[33] EQ-5D was found to be responsive to change, with moderate effect sizes at 4 and 12 months (−0.44 and −0.53, respectively).[33] We will value the EQ-5D-5L in our primary analysis in line with NICE recommendations at the time of analysis.[3 34] Quality-adjusted life years (estimated using EQ-5D-5L[32 33]) for the study period will be estimated using linear interpolation and area under the curve with and without baseline adjustment.[35] Clinical measures were found to be more responsive to change than the generic measures (shown by larger effect sizes), and combination of generic preference-based measures with the use of disease specific measure was concluded to be desirable.[33] The primary economic evaluation will be an incremental cost utility analysis to enable the cost effectiveness to be compared across a range of health conditions and interventions such that decision makers can use the information to inform prioritisation of healthcare. A secondary cost effectiveness analysis using the disease-specific Acne-QoL will be presented as appropriate, though it should be noted that this instrument does not have utility weights available and it is unclear what incremental cost per unit of change on the Acne-QoL represents good value for money. All analyses will be conducted and presented using established methods.[28 36]

If spironolactone is not found to be clinically effective, a full economic evaluation will not be conducted. Instead, estimates of mean costs and utility per participant will be presented at the various study time points as these may be informative for other researchers undertaking future economic studies or economic modelling in this clinical area.

A detailed health economic analysis plan will be written prior to the study database being locked.

## Oversight and monitoring

The TMG includes representatives with expertise in dermatology, primary care research, psychology, medical statistics and health economics, public contributors, and Southampton CTU staff involved in the day-to-day running of the study and is responsible for the oversight of the progress of the study. An independent trial steering committee and an independent data monitoring and ethics committee have been set up to monitor study progress and safety.

Data on adverse reactions will be collected at the baseline and follow-up visits, and participants will be asked to report any adverse reactions in their week 24 questionnaire. SAEs may be identified by participant report at any time directly to the hospital study team, at follow-up visits or questionnaires. Participants' GPs will be informed of their patient's participation in the study and will be asked to notify the hospital study team of any potential SAE. The study also has a UK regulatory compliant real-time SAE reporting process to identify

serious adverse reactions and suspected unexpected serious adverse reactions that could suspend or stop the study if warranted.

The Southampton CTU has undertaken a risk assessment for the study, which includes the requirements for monitoring (both central and site). The Southampton CTU undertakes a number of internal audits of its own systems and processes annually and has routine audits from both its sponsor and the independent Medicine and Healthcare Products Regulatory Authority every 2–3 years.

## DISCUSSION

This is the first adequately powered pragmatic, randomised trial investigating the effect of spironolactone on acne in adult women in comparison to a matched placebo. If found to be clinically effective and cost effective, use of spironolactone will likely become the new standard of care in addition to topical treatments potentially reducing antibiotic use for women requiring systemic therapy.

Respondents to a survey of people with experience with acne reported that they would be unwilling to be recruited to a study where they remained blinded to the treatment allocation for 52 weeks, due to concerns about potential worsening of acne over this time. Therefore, the study was designed as blinded treatment phase of 24 weeks with an observational follow-up period for up to 6 months after. Use of acne treatments, such as oral isotretinoin or antibiotics, between week 24 and up to week 52 are carefully recorded as differences between groups and would be important in interpreting week 52 outcomes. Use of topical treatments is allowed in both groups during the 24-week treatment phase as (1) women with moderate–severe acne may be unwilling to be randomised to placebo alone, and (2) recruiting women with moderate–severe acne to a placebo-controlled study with no effective treatment for 12 weeks in the control arm may risk worsening of acne and possible scarring.

Although others are seeking to evaluate the role of spironolactone in comparison with oral tetracyclines,[37] this is the largest study to date to inform clinical practice over the effectiveness of spironolactone as an alternative treatment for acne in adult women.

### Author affiliations
[1]Southampton Clinical Trials Unit, University of Southampton, Southampton, UK
[2]Primary Care Research Centre, Faculty of Medicine, School of Primary Care, Population Sciences and Medical Education, University of Southampton, Southampton, UK
[3]PPI representative, Wickham, UK
[4]Acne Support, PPI representative, Cambridgeshire, UK
[5]School of Medicine, Centre of Evidence Based Dermatology, University of Nottingham, Nottingham, UK
[6]Health Economics Group, Norwich Medical School, University of East Anglia, Norwich, UK
[7]Population Health Sciences, University of Bristol Faculty of Health Sciences, Bristol, UK
[8]Hull York Medical School, University of York, York, UK

**Acknowledgements** The authors thank the virtual patient panel (brought together by NIHR INVOLVE) for their advice on the study design. The study was developed with support from the UK Dermatology Clinical Trials Network (UK DCTN). The UK DCTN is grateful to the British Association of Dermatologists and the University of Nottingham for financial support of the Network. UK DCTN conducted surveys among patients with acne and health professionals managing acne to inform the study design, promoting the study and identifying and advising on potential hospital sites delivering the study. The authors also thank Wessex CRN for funding the initial social media advertising campaign; Jessica Boxall and Liz Allaway for management of the study social media accounts as well as running and coordinating the social media adverts; hospital dermatology centres recruiting for the study: Queen Elizabeth Hospital, Birmingham; Bristol Royal Infirmary Dermatology Centre, Bristol; University Hospital of Wales, Cardiff; General Hospital, Epsom; District Hospital, Harrogate; St Mary's Hospital (Imperial College NHS Healthcare Trust), London; Queen's Medical Centre, Nottingham; General Hospital, Poole; St Mary's General Hospital Dermatology Centre, Portsmouth; Swansea Bay University Health Board, Swansea; participant identification centres (PICs) for searching their patient lists and mail outs and clinical research networks for helping to identify potential PICs.

**Contributors** Research funding was obtained by MS, AML, NF, GG, PL, IM, JN, MJR, TS, IS, LS, BS, KT and KST. All authors contributed to the development of the protocol and to the management of the study. SR leads on the day-to-day management of the study, overseen by ZE, MS, JN and BS. This paper was drafted by FC and SR with contributions from LD, MS and all authors. All authors read and approved the final manuscript.

**Funding** This project is funded by the National Institute for Health Research (NIHR) Health Technology Assessment (HTA) programme (grant reference number 16/13/02) and supported by NIHR CTU support funding at Southampton CTU. The views expressed are those of the authors and not necessarily those of the NIHR or the Department of Health and Social Care. The NIHR HTA funder will play no role in the execution, analysis, interpretation of data or study publication. The study is registered on the UK NIHR study portfolio, meaning there are research nurses based at UK hospitals who help in screening potential patients to identify those eligible for the study. Southampton CTU, an NIHR CTU support funded UK Clinical Research Collaboration registered CTU, is coordinating the study. University of Southampton is the sponsor for the study.

**Competing interests** None declared.

**Patient consent for publication** Not required.

**Ethics approval** The trial received favourable ethical opinion from Wales Research Ethics Committee (18/WA/0420) and has Health Research Authority approval (IRAS 246637).

**Provenance and peer review** Not commissioned; externally peer reviewed.

**ORCID iDs**
Susanne Renz http://orcid.org/0000-0003-0241-7240
Ingrid Muller http://orcid.org/0000-0001-9341-6133
Kim Suzanne Thomas http://orcid.org/0000-0001-7785-7465
Matthew J Ridd http://orcid.org/0000-0002-7954-8823
Nick Francis http://orcid.org/0000-0001-8939-7312
Paul Little http://orcid.org/0000-0003-3664-1873
Miriam Santer http://orcid.org/0000-0001-7264-5260

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
