## [Reviewer comments · BMJ Open]

ARTICLE DETAILS

TITLE (PROVISIONAL)	Spironolactone for Adult Female Acne (SAFA): A double-blind, placebo-controlled, phase III randomised study of spironolactone as systemic therapy for acne in adult women: study protocol
AUTHORS	Renz, Susanne; Chinnery, Fay; Stuart, Beth; Day, Laura; Muller, Ingrid; Soulsby, Irene; Nuttall, Jacqui; Thomas, Karen; Thomas, Kim; Sach, Tracey; Stanton, Louise; Ridd, Matthew; Francis, Nick; Little, Paul; Eminton, Zina; Griffiths, Gareth; Layton, Alison M; Santer, Miriam

VERSION 1 – REVIEW

REVIEWER	Kalabalik-Hoganson, Julie Fairleigh Dickinson University School of Pharmacy and Health Sciences, Pharmacy Practice
REVIEW RETURNED	18-Jun-2021

GENERAL COMMENTS	Thank you for the opportunity to review this well-written and well-designed study. Considerations for the authors are below. Page 3 of 24, line 84, consider substituting co-cyprindiol with Ethinylestradiol/cyproterone acetate Page 4 of 24, line 85, consider discussing the use of isotretinoin for cystic, nodular, recalcitrant acne Page 4 of 24, line 85, consider discussing the early use of isotretinoin and its ability to cause remission Page 4 of 24, line 87, consider also adding the iPLEDGE program enrollment required with isotretinoin treatment Page 4 of 24, line 97, consider expanding your discussion of previously published studies on the use of spironolactone in adult female patients as there have been several retrospective studies conducted in the past 3 years that are not included in this section of the background Page 5 of 24, line 116, the severity of acne in included patients and determination of moderate to severe acne is left to the participant and study clinician, however there are acne severity grading taxonomies such as the FDA Investigator Global Assessment 2005 and the European Union Guidelines Clinical Classification Consider adding hypersensitivity to spironolactone as exclusion criteria Data management – please explain how participant identifiers will be protected
---

	Please describe if subjects will be permitted to take supplements, vitamins, herbs for the management of acne and how this information will be collected throughout the study Please describe if subjects with PCOS or hirsutism will be included in the study Please describe if there are any incentives offered to subjects for participating Please explain if subjects will be informed of the adverse effect of spironolactone, specifically irregular menses Please discuss the impact of high potassium food intake on potassium levels in subjects taking spironolactone and how subjects will be advised to avoid excessive potassium intake from food sources.
--	--

REVIEWER	Costa, Caroline
REVIEW RETURNED	Universidade Federal do Piaui 23-Jun-2021

GENERAL COMMENTS	Comments related to the Review Checklist Congratulations to the authors on the excellent and important project, SAFA trial. Item 2. Is the abstract accurate, balanced and complete? A clear and direct sentence to describe the objective of the study is missing in the end of Introduction section/Abstract. Other Comments Methods 1. In Eligibility/Inclusion criteria (line 150), the sentence “which has not been validated in other languages” should be reviewed. Please, see the following references: - Tan J, O'Toole A, Zhang X, Dreno B, Poulin Y. Cultural and linguistic validation of acne-QoL in French. J Eur Acad Dermatol Venereol. 2012;26(10):1310-4. doi: 10.1111/j.1468-3083.2011.04193.x. - Kamamoto C de S, Hassun KM, Bagatin E, Tomimori J. Acne-specific quality of life questionnaire (Acne-QoL): translation, cultural adaptation and validation into Brazilian-Portuguese language. An Bras Dermatol. 2014;89(1):83-90. doi: 10.1590/abd1806-4841.20142172.
---

VERSION 1 – AUTHOR RESPONSE

Reviewer: 1

1. Page 3 of 24, line 84, consider substituting co-cyprindiol with Ethinylestradiol/cyproterone acetate
We have changed this to, "ethinylestradiol/cyproterone acetate (co-cyprindiol)" throughout the manuscript.

2. Page 4 of 24, line 85, consider discussing the use of isotretinoin for cystic, nodular, recalcitrant acne

We have changed, "In the UK, oral isotretinoin can be used under the supervision of a dermatologist for indications including severe acne or acne that has not responded to an adequate course of a systemic antibiotics." To "In the UK, oral isotretinoin can be used under the supervision of a dermatologist for indications including severe, cystic, nodular or recalcitrant acne." Now on page 3 of 24, lines 80-82

3. Page 4 of 24, line 85, consider discussing the early use of isotretinoin and its ability to cause remission
Isotretinoin is not the main focus of this paper and we feel there is insufficient space to fully discuss this point.

4. Page 4 of 24, line 87, consider also adding the iPLEDGE program enrollment required with isotretinoin treatment

We have changed, "Oral isotretinoin is highly effective, but is contraindicated for some and is teratogenic." To "Oral isotretinoin is highly effective, but is not suitable for all patients and is teratogenic, therefore needing careful pregnancy prevention management." Now on page 3 of 24, lines 82-83.

5. Page 4 of 24, line 97, consider expanding your discussion of previously published studies on the use of spironolactone in adult female patients as there have been several retrospective studies conducted in the past 3 years that are not included in this section of the background

We believe that two of the retrospective studies the reviewer refers to are discussed in the following paragraph (Page 4 of 24, lines 93-97). We have added the wording "A further database study has shown that spironolactone may have superior drug usage survival compared to oral antibiotics for women with acne, giving a suggestion of greater perceived effectiveness and tolerability" with reference Barbieri JS, Choi JK, James WD, Margolis DJ. Real-world drug usage survival of spironolactone versus oral antibiotics for the management of female patients with acne. Journal of the American Academy of Dermatology. 2019 Sep;81(3):848. Now on page 4 of 24, lines 97-99.

6. Page 5 of 24, line 116, the severity of acne in included patients and determination of moderate to severe acne is left to the participant and study clinician, however there are acne severity grading taxonomies such as the FDA Investigator Global Assessment 2005 and the European Union Guidelines Clinical Classification

The study clinician is using the Investigator's Global Assessment to grade the trial participant's acne at baseline (in order to confirm acne severity for eligibility) and at the two following visits (in order to assess improvement compared to baseline). The Investigator's Global Assessment is a secondary outcome, as explained on page 8 of 24, lines 223-2231. The primary outcome is acne-related quality of life, as explained on page 5 of 24 lines 121-122 and on page 8 of 24, lines 214-222.

7. Consider adding hypersensitivity to spironolactone as exclusion criteria

Given that the mechanism and duration of action of spironolactone is not known, we excluded women who had ever used spironolactone, thus also excluding women with a hypersensitivity to spironolactone. In addition, recruitment for the study is already underway, so it is not possible to amend the exclusion criteria at this point.

8. Data management – please explain how participant identifiers will be protected

We have added "The site retains a participant identification code list which is only available to site staff and stored in a secure location at site" on page 10 of 24 lines 263-264

9. Please describe if subjects will be permitted to take supplements, vitamins, herbs for the management of acne and how this information will be collected throughout the study

Trial participants are allowed to take taking supplements, vitamins and herbs for the management of acne

whilst taking part in the pragmatic trial. Trial participants are at each time point asked which topical or oral treatments were used in the period since the previous time point (if any) and have the option to add more information of the treatments for their acne in a free text box on the participant questionnaires. This is discussed on page 8 of 24 lines 224-227.

10. Please describe if subjects with PCOS or hirsutism will be included in the study

Patients with PCOS and/or hirsutism will be included in the study. At the baseline visit, the trial participant is asked questions to establish if a previous diagnosis of PCOS has been confirmed or whether they are experiencing symptoms consistent with PCOS. There will be a planned subgroup analysis to assess whether the treatment effect differs by whether [participants have symptoms consistent with PCOS/were diagnosed with PCOS in the past. This is discussed on page 11 of 24 lines 308-310.

11. Please describe if there are any incentives offered to subjects for participating

Trial participants receive shopping vouchers at baseline, 6 and 12 weeks (total £40). This was added on page 5 of 24 lines 124-125 and is also detailed in Figure 1.

12. Please explain if subjects will be informed of the adverse effect of spironolactone, specifically irregular menses

All known adverse effects of spironolactone are detailed in the Participant Information Sheet. Trial participants have the opportunity to discuss the trial with the site team and ask questions before consenting to enter the trial. This is discussed on page 7 of 24 lines 180-182.

13. Please discuss the impact of high potassium food intake on potassium levels in subjects taking spironolactone and how subjects will be advised to avoid excessive potassium intake from food sources. Serum potassium levels and renal function are measured at baseline to ensure enrolled patients have normal renal function. We did not provide participants with advice to avoid high potassium foods as the Summary of Product Characteristics indicates that the avoidance of excessive potassium intake from food sources is not necessary in a young healthy population. A large study identified that the rate of hyperkalemia in young healthy women taking spironolactone is equivalent to the baseline rate of hyperkalemia in this population (Plovanich M, Weng QY, Mostaghimi A. Low Usefulness of Potassium Monitoring Among Healthy Young Women Taking Spironolactone for Acne. *JAMA Dermatol.* 2015 Sep;151(9):941-4. doi: 10.1001/jamadermatol.2015.34. PMID: 25796182)

Reviewer: 2

1. Item 2. Is the abstract accurate, balanced and complete?

We have taken care to ensure that all requisite items are included in the abstract and that it balanced and accurate.

2. A clear and direct sentence to describe the objective of the study is missing in the end of Introduction section/Abstract.

We have added a sentence describing the objective of the study at the end of the introduction and in the abstract.

3. Methods: 1. In Eligibility/Inclusion criteria (line 150), the sentence “which has not been validated in other languages” should be reviewed. Please, see the following references: - Tan J, O'Toole A, Zhang X, Dreno B, Poulin Y. Cultural and linguistic validation of acne-QoL in French. *J Eur Acad Dermatol Venereol.* 2012;26(10):1310-4. doi: 10.1111/j.1468-3083.2011.04193.x.
- Kamamoto C de S, Hassun KM, Bagatin E, Tomimori J. Acne-specific quality of life questionnaire (Acne-

QoL): translation, cultural adaptation and validation into Brazilian-Portuguese language. An Bras Dermatol. 2014;89(1):83-90. doi: 10.1590/abd1806-4841.20142172

VERSION 2 – REVIEW

REVIEWER	Kalabalik-Hoganson, Julie Fairleigh Dickinson University School of Pharmacy and Health Sciences, Pharmacy Practice
REVIEW RETURNED	19-Jul-2021
GENERAL COMMENTS	Excellent study, looking forward to the results, thank you for the opportunity to review.